# Validation of the Polish Version of Knee Outcome Survey Activities of the Daily Living Scale in a Group of Patients after Arthroscopic Anterior Cruciate Ligament Reconstruction

**DOI:** 10.3390/jcm12134317

**Published:** 2023-06-27

**Authors:** Magdalena Szczepanik, Jarosław Jabłoński, Agnieszka Bejer, Katarzyna Bazarnik-Mucha, Joanna Majewska, Sławomir Snela, Daniel Szymczyk

**Affiliations:** 1Institute of Health Sciences, College of Medical Sciences, University of Rzeszow, Rejtana 16C, 35-959 Rzeszow, Poland; abejer@ur.edu.pl (A.B.);; 2Institute of Medical Sciences, College of Medical Sciences, University of Rzeszow, Rejtana 16C, 35-959 Rzeszow, Poland; 3Orthopaedics and Traumatology Clinic for Adults, Clinical Hospital No. 2, 35-301 Rzeszow, Poland

**Keywords:** KOS-ADLS, ACL reconstruction, psychometrics, validation

## Abstract

Background: The study aimed to assess the reliability, validity, and responsiveness of the Polish version of Knee Outcome Survey Activities of the Daily Living Scale (KOS-ADLS) in a group of patients after arthroscopic reconstruction of the anterior cruciate ligament (ACL). Methods: The study was a longitudinal study with repeated measures. One hundred and twelve subjects who qualified for arthroscopic ACL reconstruction (mean age = 31.8 years) were initially enrolled in this study. The Polish version of KOS-ADLS and Short Form-36 v. 2.0 (SF-36) were used. Results: The Polish version of KOS-ADLS in subjects after ACL rupture demonstrated excellent internal consistency (Cronbach’s alpha for KOS-ADLS- total = 0.91), and test–retest reliability using the intraclass correlation coefficient (ICC-total = 0.98). The standard error of measurement (SEM) value was 0.81 and the minimal detectable change (MDC) was 2.23 for KOS-ADLS-total. The validity analysis showed a moderate and low correlation between KOS-ADLS and different domains of SF-36 from r = 0.354 between KOS-ADLS activity and the physical component scale (PCS) of SF-36: to r = 0.206 between KOS-ADLS activity and the mental component scale (MCS) of SF-36. Conclusions: The Polish version of KOS-ADLS turned out to be a reliable, valid and responsive self-reported outcome measure, allowing for the self-assessment of symptoms and function related to the knee joint impairment after ACL reconstruction. Therefore, the scale can be applied in clinical practice and research.

## 1. Introduction

Knee joint conditions are one of the major public health concerns, resulting in functional problems and a lower quality of life [1]. Tears of the anterior cruciate ligament (ACL) belong to the most frequent injuries of the knee joint, particularly in younger adults (aged 20–29 years) participating in sports that involve pivoting, jumping, acceleration and deceleration, changing direction, as well as direct contact between athletes [2,3]. Non-operated ACL injury may lead to progressing functional instability, secondary meniscal tears, as well as the possible early onset of knee osteoarthritis. Therefore, ACL reconstruction is used as the first choice treatment for ACL tears to restore proper stability and biomechanics of the knee joint, enabling a safe return to sport and physical activity and improving the subject‘s quality of life [4]. One of the main objectives of ACL reconstruction surgery is to return to sport and performance at the pre-injury level, although a high percentage of patients never achieve that goal. Some patients can still present functional limitations in terms of physical, social and psychological functioning [5]. The reasons for that situation are multifactorial, covering many areas related to the surgery and rehabilitation process, which may also indicate the need to promote some educational programs for training orthopedic surgeons [6,7]. Validated clinical and scientific tools need to be used to evaluate the post-injury status and clinical outcomes in these patients. Knee joint-specific, patient-reported measures or scales are designed to focus on clinical symptoms and the patients’ subjective symptoms to evaluate the impact of the injury on knee function and overall quality of life. This may be translated into an improved diagnosis, more detailed evaluation of the injured knee functional status, and more specific and better treatment options. These tools need to be simple to apply and use for patients having various knee-related conditions [8]. A range of joint- and disease-specific disability scales were developed for patients with knee disorders [9]. Knee Outcome Survey Activities of the Daily Living Scale (KOS-ADLS) was originally developed and validated in 1998 in the USA by Irrgang [10]. It includes 14 items arranged in as a self-administered questionnaire with six responses available for each item. The initial six items are intended to assess the knee joint-related symptoms, while the eight remaining items address the knee functional performance [10]. Most of the existing, patient self-administered questionnaires are in English. The application of these tools in non-English speaking subjects and different cultural areas requires their translation, cross-cultural adaptation and validation [11]. Such an adaptation process needs to be performed according to the methodology proposed in the scientific literature [12,13]. KOS-ADLS has been already validated in countries such as Portugal, Saudi Arabia, Kuwait, Germany, Turkey, Greece, Brazil, China, Iceland, Canada, and Iran, as well as Poland, in group of patients after total knee replacement (TKR) [11,14,15,16,17,18,19,20,21,22,23,24]. The aim of this study was to evaluate the reliability, validity, and responsiveness of the Polish version of KOS-ADLS in a group of patients after arthroscopic ACL reconstruction.

## 2. Materials and Methods

### 2.1. Linguistic and Cross-Cultural Validation Process

The cross-cultural adaptation process of KOS-ADLS in our study was in line with the standard guidelines described in the previous study, performed in patients with total knee replacement [24].

### 2.2. Study Design and Population

Patients that underwent primary ACL reconstruction or combined ACL reconstruction with meniscal repair or resection took part in this study. All patients who were qualified for this surgery between February 2018 and May 2021 in the Orthopaedics and Traumatology Clinic for Adults, Clinical Hospital No. 2 in Rzeszow (Poland) were enrolled. The inclusion criteria were: adults over 18 years old; Polish native speakers; and patient’s written informed consent. The exclusion criteria were as follows: previous injuries (fracture, location, sub-location) and surgeries of the knee and lower limb; previous ACL injury in both knees; and other concurrent neurological disorders (sclerosis multiplex, polyneuropathy) and musculoskeletal disorders (osteoarthritis, rheumatoid arthritis) relating to the lower limbs, which may have had an impact on patients’ medical condition and symptoms.

### 2.3. Ethics

The study was approved by the Bioethics Committee of the Medical College of the University of Rzeszow (Resolution No.3.01.2018). All participants were informed about the procedures of these studies and signed written informed consent.

### 2.4. Measures

#### 2.4.1. Knee Outcome Survey- Activities Daily Living Scale

The KOS-ADLS is a 14-item, self-reported questionnaire assessing symptoms and functional limitations related to the knee disorders. Six items evaluate knee symptoms (KOS-ADLS symptoms): pain, stiffness, swelling, instability, weakness, limping and eight remaining items evaluate functional limitations: walking, stairs ascent/descent, standing, kneeling, squatting and rising from a sitting position (KOS-ADLS activity of daily living). Every item is scored on 0–5 points Likert scale. The final KOS-ADLS score is determined by dividing the KOS-ADLS patient‘s total score by 70 and then multiplying the result by 100, where 100 indicates the lack of symptoms and functional limitations [10,15].

#### 2.4.2. The SF-36 Health Survey 2.0 (SF-36 2.0)

The SF-36 2.0, being a generic Health Related Quality of Life (HRQOL) questionnaire, includes 36 items measuring eight domains: physical functioning (PF), physical role functioning (PRF), bodily pain (BP), general health (GH), vitality (VT), social role functioning (SRF), emotional role functioning (ERF) and mental health (MH). A score from 0 (worst possible health status) to 100 (best possible health status) is calculated independently for every domain. The first four domains refer to physical component scale (PCS) and the next four refer to the mental component scale (MCS) [20,25].

### 2.5. Study Procedure

All patients qualified for this study completed the questionnaires four times. The first evaluation took place one to two weeks before surgery and consisted of completing the Polish version of KOS-ADLS and SF-36. The second evaluation took place three months postoperatively, and the patients were asked to complete the same questionnaires. The third assessment was performed six months after surgery, and the patients completed KOS-ADLS only. Additionally, one week later, the subjects completed the KOS-ADLS questionnaire for the last time; this was the fourth evaluation (test–retest). During the baseline evaluation, the patients filled out the paper version of the questionnaires in the presence of the researchers, who explained any potential doubts. In the next stages of this study, the questionnaires were sent to the subjects by e-mail. Prior to qualification for this study, all subjects were informed about its procedures.

### 2.6. Statistical Analysis

All statistical analysis were performed with the R software version 4.1.2. [R Core Team (2021). R: A language and environment for statistical computing. R Foundation for Statistical Computing, Vienna, Austria. URL https://www.R-project.org/ (assessed on 29 April 2023). The statistical significance level was assumed at *p* < 0.05. For the purpose of statistical analysis, the descriptive statistics were applied (mean value, standard deviation, median and quartiles). The Wilcoxon signed-rank test was performed to compare changes between the first and the second evaluation. Floor and ceiling effect were assumed to be present in case the percentage of the lowest or highest scores was greater than 15% [26].

#### 2.6.1. Sample Size

Sample size was predefined according to the Altman guidelines, in which a study group should consist of at least 50 subjects [27]. Additionally, post hoc power analysis was conducted for the intraclass correlation coefficient at 95% confidence interval (ICC).The sample size was n = 62 (the number of subjects who participated in every stages of this six months follow-up study). Null hypothesis of ICC = 0.7 was assumed [26]. This is the value of ICC commonly used as a cut-off point for satisfactory reliability level. For each of the KOS-ADLS subscales, the calculated power of rejecting null hypothesis was over 0.99, which indicates that the sample size in our study (n = 62) was adequate.

#### 2.6.2. Reliability

Internal consistency

Internal consistency was calculated using Cronbach’s alpha coefficient, based on the data from baseline evaluation (N = 112). An alpha value ≥ 0.7 is described to be satisfactory [26,28].

Reliability (test–retest)

To evaluate reliability, an intraclass correlation coefficient (ICC) with a 95% confidence interval (CI) was used. ICC values of 0.90 and above are considered an excellent correlation, 0.75–0.90 are considered good, 0.50–0.74 moderate and below 0.50 poor [29]. Additionally, the standard error of measurement (SEM) and minimum detectable change at 95% CI (MDC) were calculated [24]. The ICC, MDC and SEM were calculated from test–retest data obtained at the six months follow up (third and fourth evaluation). The mean time, in which the patients sent back the KOS-ADLS questionnaire, was five days (ranging from one to seven days). Sixty-two subjects participated in this stage of the study.

#### 2.6.3. Construct Validity

The construct validity was calculated from data from baseline evaluation (N = 112).

Hypotheses testing

To assess of the construct validity of KOS-ADLS, the Spearman’s rank correlation coefficient (SCC) was used. For this purpose, correlation between the KOS-ADLS- symptoms, KOS-ADLS- activity of daily living and KOS-ADLS- total and specific domain of SF-36 were calculated. The levels of agreement were defined as ‘Excellent’ for coefficients ±0.90–±1, ‘High’ for ±0.50–±0.89, ‘Moderate’ for ±0.30–±0.49 and ‘Low’ for <±0.29 [30]. We initially hypothesized that the correlation between KOS-ADLS- symptoms, KOS-ADLS- activity of daily living and KOS-ADLS- total and the SF-36: total of Physical component, would be moderate or strong and positive (hypotheses 1–3). Correlation between KOS-ADLS- symptoms, KOS-ADLS- activity of daily living and KOS-ADLS- total and the SF-36: total of Mental component, would be moderate or low and positive (hypotheses 4–6).

In order to evaluate the construct validity of the KOS-ADLS, the “known group” analysis also were used. Patients were divided according to the SF-36 domain BP for two groups. The first group included patients who described their level of pain as a lack of pain, very mild or mild pain, while the second group were the subjects who described their level of pain as moderate, severe or very severe. The Mann–Whitney test was used for hypothesis testing. We hypothesized that: the questionnaire KOS-ADLS- total can distinguish between two groups of patients which have different levels of pain (7-th hypothesis).

#### 2.6.4. Responsiveness

According to the Husted recommendation [31], we evaluated internal and external responsiveness.

To evaluate the internal responsiveness, Standard Effect Size (ES) and Standardized Response Mean (SRM) were calculated based on data obtained from baseline and the three month follow-up evaluation [24,31]. The value of ES and SRM ≥ 0.8 are defined as a large responsiveness [24].To evaluate external responsiveness, we compared changes in the KOS-ADLS and SF-36 between the data obtained from baseline and the three month follow-up evaluation. We used Spearman’s rank correlation coefficient (SCC). We assumed that the changes in the KOS-ADLS symptoms, KOS-ADLS activity of daily living and KOS-ADLS total would be moderate or strongly correlated with changes in the total score of the physical component scale (PCS) of SF-36, and would show a positive correlation. We also expected that changes in the KOS-ADLS symptoms, KOS-ADLS activity of daily living and KOS-ADLS total would present a moderate or low correlation with changes in the total score on the mental component scale (MCS) of SF-36, and would be positive.

The responsiveness was calculated from data obtained from 88 patients participating at these stages of the study.

## 3. Results

### 3.1. Description of the Study Population

One hundred and twelve patients meeting the inclusion criteria participated in the baseline evaluation. A total of 88 subjects participated in the three month follow-up evaluation. Two patients had re-injury of the same knee, and eighteen patients did not send back the questionnaires, while four patients sent back incomplete questionnaires. In the test–retest evaluation (the third 6 month follow-up evaluation), 70 patients sent back the KOS-ADLS questionnaire, while 62 patients participated in the last stage of this study (8 patients did not send back KOS-ADLS). Patient characteristics are presented in Table 1.

### 3.2. Validation Study

Table 2 includes the data from the baseline evaluation. The floor and ceiling effect for KOS-ADLS symptoms, KOS-ADLS activity of daily living and KOS-ADLS total was not present.

#### 3.2.1. Reliability

##### Internal Consistency, Test–Retest Reliability and Measurement Error

Internal consistency was excellent with a Cronbach’s alpha value of 0.91 for the total score of KOS-ADLS. The reliability was also excellent (total ICC = 0.98), SEM value was 0.81 and MDC 2.23 for the total score of KOS-ADLS (Table 3).

#### 3.2.2. Construct Validity

##### Hypotheses Testing

The KOS-ADLS activity and KOS-ADLS total were significantly correlated with SF-36 PCS (and the value of correlation was moderate), while the correlation between KOS-ADLS symptoms and SF-36 PCS was low. As it has been hypothesized, the correlations between KOS-ADLS symptoms, KOS-ADLS activity, KOS-ADLS total and SF-36 MCS were significant but low (Table 4).

Statistically significant differences between two groups with different level of pain determined in SF-36 BP showed an appropriate discrimination power of the KOS-ADLS (*p* = 0.004) (Table 5).

To assess construct validity, six out of seven—‘a priori’ assumed hypothesis (85.7%)—were confirmed. According to Terwee’s et al., guidelines concerning the quality criteria of health status questionnaires indicates a high-construct validity of KOS-ADLS [26].

#### 3.2.3. Responsiveness

##### Internal Responsiveness

Between baseline and the 3 month follow-up evaluation, statistically significant changes were seen in all subscales and the total result of KOS-ADLS. Values of ES and SRM showed excellent responsiveness for all subscales and the total result of KOS-ADLS (Table 6).

##### External Responsiveness

According to our hypothesis, moderate and statistically significant correlation were observed between changes in KOS-ADLS symptoms, KOS-ADLS activity of daily living, KOS-ADLS total and SF-36 PCS. Moreover, a correlation between changes for both subscales of KOS-ADLS, overall KOS-ADLS and SF-36 MCS were moderate and statistically significant (Table 7).

## 4. Discussion

The original KOS-ADLS scale is a valid and reliable self-reported questionnaire for the evaluation of symptoms and functional limitations related to the knee disorders [10,11,14,15,16,17,18,19,20,21,22,23,24]. The use of this scale has been also widely reported in studies in group of patients with different knee problems [32,33,34,35]. KOS-ADLS was translated and adapted into Polish and validated in patients after TKR [24]. Therefore, we decided to assess the clinimetric properties, and the internal consistency, validity and reliability of the Polish version of KOS-ADLS in patients after arthroscopic ACL reconstruction. To the best of our knowledge, this study is the first to concern the validation of KOS-ADLS in a large population of patients after arthroscopic ACL reconstruction.

To assess internal consistency of the KOS-ADLS in our group of patients, we used Cronbach’s alpha coefficient. In this study, the internal consistency was excellent with a Cronbach’s alpha of 0.91 for the total score of KOS-ADLS. It is commonly assumed that Cronbach’s alpha values between 0.70 and 0.90 indicate good internal consistency [26]. The Cronbach’s alpha value obtained in our study for the KOS-ADLS was approximately at the same level as those reported for the original KOS-ADLS [10], and the other cross-culturally adapted versions (from 0.87 to 0.99) [11,14,15,16,17,18,19,20,21,22,23,24]. In our previous study, which concerned the Polish version of the KOS-ADLS in patients qualified for TKR in the preoperative study, the internal consistency was poor at 0.68, while the postoperative examinations reached 0.86 [24]. In our study, the reliability was also excellent (total ICC = 0.983), the SEM value was 0.805 and MDC was 2.231 for the total score of KOS-ADLS. Our results are similar to the original version created by Irgang (ICC 0.97) [10], and the results obtained in the studies using KOS-ADLS conducted in other countries are: (Portugal- ICC = 0.97, Kuwait (Arabic version)- ICC= 0.97, Germany- ICC = 0.97, Turkey- ICC = 0.99, Greece- ICC = 0.97, China- ICC = 0.94, Iceland- ICC = 0.95, Canada (French version)- ICC = 0.92) and Iran (Persian version)- ICC = 0.89) for the total score of KOS-ADLS [11,15,16,17,18,20,21,22,23]. Comparative with our previous study using the Polish version of KOS-ADLS in patients after TKR (ICC ranged from 0.88 to 0.89, with SEM 2.68 and MDC 7.43), we obtained significantly better parameters, despite a longer interval between the test and retest evaluation [24]. Marx et al. assessed the reliability, validity, and responsiveness of four knee outcome scales, including the KOS-ADLS, in athletic patients. The KOS-ADLS was considered reliable (ICC, 0.93), valid, and responsive enough to be applied in clinical research. It had a better validity and responsiveness than the other scales (Lysholm scale, Cincinnati knee-rating system, the American Academy of Orthopaedic Surgeons sports knee-rating scale). The authors explain this fact to the clear wording of the instrument or to the fact that it evaluates a wide range of symptoms and disabilities, compared with the other questionnaires. They suggest that KOS-ADLS can be used for various types of knee conditions and knee-related sports injuries, including anterior cruciate ligament injuries [36].

In our study, the floor and ceiling effect for KOS-ADLS symptoms, KOS-ADLS activity of daily living and the KOS-ADLS total was not observed, indicating that the study group (sample) were homogenous in symptoms and functional limitations. Marx et al. also did not observe ceiling or floor effects for KOS-ADLS in athletic patients [36]. In the Greek version, in which the study group consisted of patients with various pathological disorders and types of injuries, the ceiling or floor effects were also not observed [18]. Our results were also comparable to the original KOS-ADLS, as well as the Portuguese and Arabic versions [10,11,15]. Although, Szczepanik et al. (KOS-ADLS Polish version) observed a very high level of floor/ceiling effects in the preoperative assessment of patients qualified for TKR, the floor effect was related to the item concerning squatting and kneeling, and this was mainly due to the characteristic of the study group, which consisted of patients with advanced, end-stage knee OA. They simply avoided this activity due to high-pain levels and muscle weakness. Postoperatively, the floor effect was observed in the question concerning kneeling and a ceiling effect in the question of giving way. It could be explained by a significant fear of kneeling in patients after TKR [24].

The construct validity of the original KOS-ADLS demonstrated fair to good correlations with the Lysholm Knee Scale (r = 0.78–0.86) and the global rating of function (r = 0.66–0.75) [10]. The SF-36 questionnaire was applied to validate the Arabic, Portuguese, Brazilian, and Chinese versions of KOS-ADLS [11,15,19,20]. The level of correlation depended on which subscale of SF-36 was correlated with KOS-ADL. In our study, as we have hypothesized, the KOS-ADLS activity and KOS-ADLS total significantly correlated with SF-36 PCS, and the value of correlation was moderate (KOS-ADLS activity r = 0.354, KOS-ADLS total r = 0.344). Whereas, the correlation between KOS-ADLS symptoms and SF-36 PCS was low (r = 0.24). On the other hand, the correlation between KOS-ADLS symptoms, KOS-ADLS activity, the KOS-ADLS total and SF-36 MCS was statistically significant but low (r = 0.233, r = 0.206, r = 0.241, respectively). In the Chinese version, the correlations ranged from poor (subscale “mental health”) to very good (subscale “physical function”) [20]. Marx et al. observed good to excellent correlations with various knee outcome measures (r = 0.68– 0.85) and the SF-36 physical component scale (r = 0.77) [36]. In the Polish version of the KOS-ADLS in a group of patients after TKR, the strongest correlation was noted between the total score of KOS-ADLS and the subscale “activities in daily living” of KOOS (r = 0.63) [24].

In our study, we confirmed high levels of responsiveness of KOS-ADLS (ES = 1.41, SRM = 1.5) in patients after arthroscopic ACL reconstruction. In our previous validation study, in patients who underwent TKR, we also reported high ES = 4.76 and SRM = 3.18 in the 6 months follow up [24]. Irgang et al., in their original study, obtained an ES level of 0.44 after one week of physical therapy, 0.94 after four weeks and 1.94 after eight weeks of physical therapy [10]. The responsiveness of KOS-ADLS was also evaluated in the Arabic validation study (ES = 1.12, SRM = 1.09 after four weeks of physical therapy), in the Portuguese validation study, the Greek version (ES = 1.31, SRM = 1.64 after 3.9 ± 2.6 weeks of physical therapy), the French version (SRM= 1.41, Clinically Important Difference (CID) = 13.6 after four weeks of physical therapy), and the Chinese validation study (ES = 0.82, SRM = 1.18 after four weeks of physical therapy) [11,15,18,20,22]. Marx et al. assessed the responsiveness of KOS-ADLS at a minimum of three months after surgical or nonsurgical treatment among athletic patients with knee disorders. They reported a high SRM = 1.1 [36]. Williams et al. reported a similar and high reliability and responsiveness of KOS-ADLS (SEM = 4.52–7.6), compared to the Western Ontario and McMaster Universities Arthritis Index (WOMAC), and Lower Extremity Functional Scale (LEFS) in patients with knee OA undergoing physical therapy [37]. The comparison of reliability, responsiveness and construct validity of four questionnaires (KOS-ADLS, WOMAC, 12-item Short Form Healthy Survey (SF-12) and Oxford Knee Score (OKS)), for the assessment of patients six months after TKR, was also conducted by Impellizzeri et al. The study confirmed large SRM for all tools (KOS-ADLS SRM = 4.1) [38]. Piva et al. assessed the responsiveness of KOS-ADLS and the Numeric Pain Rating Scale (NPRS) in patients with patellofemoral pain undergoing an eight-week physical therapy program, and they reported the ES of KOS-ADLS on a level of 0.63 [39].

### 4.1. Strengths

The study’s strengths include the use of standardized methods for the assessment of the psychometric properties based on COSMIN guidelines [40]. The Polish version of KOS-ADLS questionnaire can be successfully applied in scientific research, and in clinical settings. It might be helpful for the clinician to better understand the impact of arthroscopic ACL reconstruction on the patients’ quality of life and activities of daily living. It has been also recommended for monitoring an individual patient’s progress after arthroscopy and for decision-making related to further treatment. Additionally, the questionnaire is short, easy to understand for the patients, and self-administered, so it can reduce administration burdens for the clinicians.

### 4.2. Limitations and Future Considerations

The major limitations of our study concern the study group, which included only patients after arthroscopic ACL reconstruction. A previous validation study, concerning the Polish version of KOS-ADLS, included patients qualified for Total Knee Arthroplasty due to advanced knee osteoarthritis. We assume that further stages of the validation process of KOS-ADLS should include patients qualified for other knee surgical procedures, due to different knee conditions, as well as knee-related rehabilitation process.

## 5. Conclusions

In our study, concerning the Polish version of the KOS-ADLS in patients after arthroscopic anterior cruciate ligament reconstruction, we observed reliability, validity, floor/ceiling effects, and responsiveness similar to the results of the original version. The good psychometric properties of KOS-ADLS prove that it may be applied as a standardized, patient self-reported outcome measure, and disease-specific scale in clinical practice for the evaluation of the patients’ symptoms and functions related to knee joint impairment in patients, after arthroscopic anterior cruciate ligament reconstruction.

## Figures and Tables

**Table 1 jcm-12-04317-t001:** Study population characteristics (baseline and three and six month follow-up evaluation).

Variables	Baseline Evaluation (N = 112)		3 Month Follow Up Evaluation (N = 88)		6 Month Follow Up Evaluation (Test) (N = 70)		6 Month Follow Up Evaluation (Retest) (N = 62)	
	Number	Mean (Range)	Number	Mean (Range)	Number	Mean (Range)	Number	Mean (Range)
Gender								
Male	91 (81.2)		71 (80.7)		53 (75.7)		47 (75.8)	
Female	21 (18.8)		17 (19.3)		17 (24.3)		15 (24.2)	
Age (years)		31.8 (18–53)		32.9 (19–53)		32.1 (19–53)		32.7 (20–53)
Operated knee side								
Right	63		47		38		32	
Left	49		41		32		30	
Time from injury to surgery		25.5 months (1–360)						
Type of surgery								
ACLR	36		26		22		21	
ACLR, MM or LM	40		33		27		25	
ACLR, MLM	6		6		5		4	
ACLR, MMR or LMR	20		13		9		7	
ACLR, MMR and LMR	2		2		2		1	
ACLR, MMR and LM or MM and LMR	8		8		5		4	

Abbreviations: N- number of patients; ACLR- anterior cruciate ligament reconstruction; MM- medial meniscectomy; LM- lateral meniscectomy; MLM medial and lateral meniscectomy; MMR- medial meniscus repair; LMR lateral meniscus repair.

**Table 2 jcm-12-04317-t002:** Data from baseline evaluation, floor and ceiling effect for all questionnaires (N = 112).

Parameter	Mean	SD	Median	Quartiles	Floor Effect	Ceiling Effect
KOS-ADLS- symptoms (0–100)	61.10	21.70	63.33	50.00–76.67	1.8%	0.9%
KOS–ADLS- activities of daily living (0–100)	58.65	23.91	62.50	37.50–80.00	0.0%	1.8%
KOS-ADLS- total (0–100)	59.70	21.41	61.43	44.29–75.71	0.0%	0.9%
SF-36: PCS	48.10	15.46	48.24	37.35–60.29	0.0%	0.0%
SF-36: MCS	69.68	17.50	73.21	58.93–83.93	0.0%	0.0%

Abbreviations: KOS-ADLS- Knee Outcome Survey- Activities of Daily Living SF-36: PCS- SF-36 Health Survey: physical component scale; SF-36: MCS–SF-36 Health Survey: mental component scale; SD- standard deviation.

**Table 3 jcm-12-04317-t003:** Reliability analysis: internal consistency (N = 112), test–retest reliability measurement—(N = 62).

KOS-ADLS	Cronbach’s Alpha N = 112	ICC (95%CI) N = 62	SEM N = 62	MDC N = 62
KOS-ADLS- symptoms	0.76	0.96 (0.94–0.97)	0.50	1.37
KOS-ADLS- activities of daily living	0.90	0.96 (0.96–0.98)	0.60	1.67
KOS-ADLS- total	0.91	0.98 (0.97–0.99)	0.81	2.23

Abbreviations: KOS-ADLS- Knee Outcome Survey- Activities of Daily Living Scale; ICC- intraclass correlation coefficient at 95% confidence interval; SEM- standard error of measurement; MDC- minimal detectable change.

**Table 4 jcm-12-04317-t004:** Construct-validity correlation between KOS-ADLS and selected domain of the SF-36 (N = 112).

SF-36	KOS-ADLS- Symptoms	KOS-ADLS- Activities of Daily Living	KOS-ADLS- Total
PCS	r = 0.24, *p* = 0.012	r = 0.35, *p* < 0.001	r = 0.34, *p* < 0.001
MCS	r = 0.23, *p* = 0.015	r = 0.21, *p* = 0.031	r = 0.24, *p* = 0.012

Abbreviations: SF-36: PCS- SF-36 Health Survey: physical component scale; SF-36: MCS- SF-36 Health Survey: mental component scale; KOS-ADLS- Knee Outcome Survey- Activities of Daily Living Scale, r- Spearman’s rank correlation coefficient; *p*- statistically significant *p* < 0.05.

**Table 5 jcm-12-04317-t005:** Construct validity, comparison between ‘known groups’ (N = 112).

KOS-ADLS- Total	SF-36		*p*
	Group 1 (N = 70)	Group 2 (N = 42)	
mean ± SD	63.71 ± 23.26	49.55 ± 22.58	*p* = 0.004
Median	65.00	47.50	
Quartiles	45.00–85.00	35.00–68.75	

Abbreviations: SF-36: PCS- SF-36 Health Survey: physical component scale; SF-36: MCS- SF-36 Health Survey: mental component scale; KOS-ADLS- Knee Outcome Survey- Activities of Daily Living Scale; SD- standard deviation; *p*- statistically significant *p* < 0.05.

**Table 6 jcm-12-04317-t006:** Internal responsiveness- comparison between baseline and 3 month follow-up evaluation; values of ES and SRM.

Parameter		Baseline (N = 112)	3 Months Follow-Up (N = 88)	*p*	ES	SRM
KOS-ADLS- symptoms	mean ± SD	61.10 ± 21.70	75.84 ± 14.35	*p* < 0.001	1.36	1.47
	Median	63.33	76.67			
	Quartiles	50.00–76.67	66.67–83.33			
KOS-ADLS- activities of daily living	mean ± SD	58.65 ± 23.91	74.95 ± 14.81	*p* < 0.001	1.31	1.40
	Median	62.50	77.50			
	Quartiles	37.50–80.00	67.5–85.00			
KOS-ADLS- total	mean ± SD	59.70 ± 21.41	75.33 ± 13.87	*p* < 0.001	1.41	1.50
	Median	61.43	77.14			
	Quartiles	44.29–75.71	68.57–85.71			

Abbreviations: KOS-ADLS- Knee Outcome Survey- Activities of Daily Living Scale; SD- standard deviation; ES- effect size, SRM- standardized response mean; *p*- statistically significant *p* < 0.05.

**Table 7 jcm-12-04317-t007:** External responsiveness, correlation between changes in KOS-ADLS and SF-36 PCS and MCS.

SF-36	KOS-ADLS Symptoms	KOS–ADLS- Activities of Daily Living	KOS-ADLS– Total
PCS	r = 0.36, *p* < 0.001	r = 0.43, *p* < 0.001	r = 0.44, *p* < 0.001
MCS	r = 0.32, *p* = 0.001	r = 0.33, *p* < 0.001	r = 0.36, *p* < 0.001

Abbreviations: SF-36: PCS- SF-36 Health Survey: physical component scale; SF-36: MCS- SF-36 Health Survey: mental component scale; KOS-ADLS-Knee Outcome Survey Activities of Daily Living Scale; r- Spearman’s rank correlation coefficient; *p*- statistically significant *p* < 0.05.

## Data Availability

The data that support the findings of this study are available from the corresponding author upon reasonable request.

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
