# Peer review of "Validation of the Polish Version of Knee Outcome Survey Activities of the Daily Living Scale in a Group of Patients after Arthroscopic Anterior Cruciate Ligament Reconstruction"

_jcm, 2023, doi:10.3390/jcm12134317_

Round 1

Reviewer 1 Report

Overall well written paper

could have included another knee specific scale for construct validity

grammar corrections line 44-49

overall comprhensibe paper

the scientific language in English could ne slightly improved

Reviewer 2 Report

Dear Author, 

I am pleased to submit to you my review of your article. 

The topic is interesting and relevant.

The article is well written, but many concerns burden it with a minor revision before it can be accepted for publication. After these minor revisions, in my opinion, it is publishable. Congratulations to the authors for their efforts.

INTRODUCTION

The Introduction is too long. It should contain information about why this study is needed, which gap in the literature you would like to fill, and the purpose and hypothesis. Please make this section shorter.

Line 38: "contact between athletes [2]" You may add this recent article (DOI): 10.1016/j.jor.2022.11.018

Line 48: "functioning [4]" You may add this recent article (DOI): 10.1007/s00590-022-03419-4

RESULTS

-The Results section is quite long. Some results are repeated in tables and figures. This is unnecessary, and repetitions never improve a manuscript, only longer. Please make sure that repetitions are kept to a minimum. 

DISCUSSION

-This section is well done and has a logical narrative

-At the end of the discussion section, add the study's strengths, essential to a high-quality written manuscript.

Please mention the clinical relevance of your work. How may this information be helpful for orthopedic surgeons?

TABLES

-Your measurement methods must be given in detail. Measurement accuracy is necessary to report. Please ensure your results are given with the same accuracy as the methods. If your methods allow one decimal, the result should also be reported with one decimal. Information about measurement accuracy is essential.

Reviewer 3 Report

Dear Authors, 

 I was pleased to review the paper entitled " Validation of Polish version of Knee Outcome Survey Activities 2 of Daily Living Scale in a group of patients after arthroscopic 3 anterior cruciate ligament reconstruction" -

- MDPI –

 The present paper is very interesting, it focuses on a relevant clinical scenario, for orthopedics, potentially influencing the surgical and clinical practice for the management of knee surgery. 

Therefore, it is my opinion that the content is original, current, and relevant. 

Thus, there are some minor remarks:

- Abstract: please clarify “POD” acronym if it is written first time

- Introduction: 

Well written, concise, clearly describes the study objectives and background

Line 67: please correct “was developed and validated in 1998 in USA in 1998 by Irrgang [10].” 

Line 68: do not start sentence with acronym.

It is important for States and Scientific Societies to standardize languages and improve surgical education (you could add from MDP DOI: 10.3390/medicina58091164 )

- Method:

Lines 140-142 please better clarify the sample size including alpha, power and a standard deviation.

- results: please correct table 1 the second line of words.

- Discussion: well written and comprehensive.

Reviewer 4 Report

Abstract
The abstract is well-structured, following the common Background-Methods-Results
-Conclusion format. The abbreviations used might discourage readers as it seems complicated.

Introduction

1. The concept of patient-reported measures or scales is mentioned twice in lines 48-54. Consolidate these mentions to make the text more concise.

2. In line 66, you mentioned that KOS-ADLS was "developed and validated in 1998 in USA in 1998". One "in 1998" should be removed

3. The phrase "A validated clinical and scientific tools need to be used" (line 47-48) should either be "Validated clinical and scientific tools need to be used" or "A validated clinical and scientific tool needs to be used."

4. Organize the introduction to move smoothly from general to specific, starting with the problem (ACL injury), then the general solution (questionnaires), and finally the specific solution (your study about KOS-ADLS).
5. Use past tense for completed studies. So in line 75, it should be "since it has been validated"

Methods

1. Row 86: "Study designed" should be "Study design."

2. Row 87-88: Repetitive phrasing, "participated took part" should be "participated" or "took part."

Results

Perfectly described. No issues found

Discussions

Very well written, well done comparison with the state of the art

1. Spelling error: 'corelated' in line 335, should be 'correlated

2. Avoid repetition in discussing Cronbach's alpha (lines 291-296)

3. Formatting error: Misspelled '0,241' in line 339, should be '0.241'.

Conclusion

They are ok.

References

They are ok

I have provided instructions in the Comments above.
